# The Sensitivity of Field Populations of *Metopolophium dirhodum* (Walker) (Hemiptera: Aphididae) to Seven Insecticides in Northern China



Peipan Gong [1,2], Xinan Li [3], Chao Wang [2], Saige Zhu [2], Qiuchi Li [2], Yunhui Zhang [2], Xiangrui Li [2], Guangkuo Li [4], Enliang Liu [5], Haifeng Gao [4,*], Xueqing Yang [1,*] and Xun Zhu [2,*]

1   College of Plant Protection, Shenyang Agricultural University, Shenyang 110866, China; gpeipan@webmail.hzau.edu.cn
2   State Key Laboratory for Biology of Plant Diseases and Insect Pests, Institute of Plant Protection, Chinese Academy of Agricultural Sciences, Beijing 100193, China; 18375611798@163.com (C.W.); zsg121011@163.com (S.Z.); liqiuchi0921@163.com (Q.L.); yhzhang@ippcaas.cn (Y.Z.); xrli@ippcaas.cn (X.L.)
3   School of Resource and Environmental Sciences, Henan Institute of Science and Technology, Xinxiang 453003, China; lixinan2019@126.com
4   Key Laboratory of Integrated Pest Management on Crop in Northwestern Oasis, Xinjiang Academy of Ag-ricultural Sciences, Institute of Plant Protection Ministry of Agriculture and Rural Affairs, Urumqi 830091, China; lgk0808@163.com
5   Research Institute of Grain Crops, Xinjiang Academy of Agricultural Sciences, Urumqi 830091, China; liuenliang_513@163.com
*   Correspondence: ghf20044666@163.com (H.G.); sling@hotmail.com (X.Y.); zhuxun@caas.cn (X.Z.)

**Abstract:** Insect pests are primarily controlled by insecticides. However, the sensitivity decreases and insecticide resistance is problematic for the effective management of agriculturally important insects, including *Metopolophium dirhodum*, which is an aphid that commonly feeds on cereals. The insecticide sensitivity status and potential resistance of *M. dirhodum* field populations remain relatively unknown. In this study, the susceptibility of 19 *M. dirhodum* populations from seven provinces in Northern China to neonicotinoids, pyrethroids, organophosphates, and a macrolide (abamectin) was determined in 2017–2019. The results indicated that two populations were highly resistant to thiamethoxam, with a relative resistance ratio (RLR) of 134.03 and 103.03, whereas one population was highly resistant to beta-cypermethrin (RLR of 121.42). On the basis of the RLR, the tested *M. dirhodum* populations ranging from susceptible to showing moderate levels of resistance to imidacloprid (RLR of 1.50 to 57.29), omethoate (RLR of 1.07 to 18.73), and abamectin (RLR of 1.10 to 25.89), but they were ranging from susceptible to showing tolerance or low levels of resistance to bifenthrin (RLR of 1.14 to 6.02) and chlorpyrifos (RLR of 1.11 to 7.59). Furthermore, a pair-wise correlation analysis revealed a significant correlation between the median lethal concentrations ($LC_{50}$) for beta-cypermethrin and thiamethoxam, reflecting the cross-resistance between these two insecticides. The data obtained in our study provide timely information about aphid insecticide sensitivity, which may be used to delay the evolution of *M. dirhodum* insecticide resistance in Northern China.

**Keywords:** *Metopolophium dirhodum*; insecticide sensitivity; neonicotinoids; pyrethroids; correlation analysis

## 1. Introduction

Aphids are important sap-feeding agricultural pests that adversely affect cereal, vegetable, and fruit crops worldwide. Approximately 2% of the Aphididae species (100 of 5000) have successfully exploited agricultural ecosystems, resulting in substantial economic losses [1]. For example, *Metopolophium dirhodum* (Walker) is a common aphid on winter cereals [2,3]. This aphid, which is native to the Holarctic region, has been the most common cereal aphid species in Europe for many years, but it is now distributed worldwide [4,5].

*Metopolophium dirhodum* serves as a vector for several viruses (e.g., Barley Yellow Dwarf Virus) that can infect cereals [6]. The economic damage caused by *M. dirhodum* is well documented in countries where it is prevalent [7–9]. *Rhopalosiphum padi* and *Sitobion avenae* are the dominant aphid populations in the wheat-growing regions of China, whereas *M. dirhodum* is mainly distributed in the cooler areas in the northwestern part of the country. Despite the narrower distribution of *M. dirhodum*, our field surveys and bioassays have revealed it is more resistant to neonicotinoid insecticides [10]. In China, *M. dirhodum* was first detected in the 1980s, but interest in this insect was limited because it was a major pest only in parts of the western wheat-growing region (e.g., Rikaze, Tibet, China) [11]. However, *M. dirhodum* has recently migrated eastward, resulting in increased crop yield reduction [12,13].

Insecticides remain an important component of many pest management programs [13,14]. Neonicotinoids, pyrethroids, organophosphates, and macrolides are important groups of synthetic insecticides that are widely used to control arthropod pests [15]. Insecticide resistance has been reported for various aphid species, including *Myzus persicae* [16,17], *Aphis fabae* [18], and *Aphis gossypii* [19]. Previous studies confirmed that field populations of two wheat aphid species, *R. padi* and *S. avenae*, have evolved varying levels of resistance to insecticides with diverse modes of action [20,21]. An insecticide resistance diagnostic kit for *Sogatella furcifera* recently developed on the basis of insecticide resistance monitoring data has enabled rapid analyses under field conditions [22].

Neonicotinoids are currently used against sap-feeding insect pests, although their usage has been banned or restricted in Europe, and restricted in some states in the USA, due to concerns about side effects on bees and other pollinators [23–25]. Furthermore, *Nilaparvata lugens* field populations are highly resistant to neonicotinoids, which has resulted in the frequent inability to prevent infestations by this insect [26]. In addition to neonicotinoid-resistant pests, disease vector insects highly resistant to pyrethroids have been reported, including *Anopheles sinensis*, *Anopheles funestus*, and *Anopheles gambiae* [27–29]. The long-term use of pyrethroids has resulted in the emergence of cotton-melon aphid (*A. gossypii*) populations highly resistant to these insecticides [30]. The organophosphate insecticides were frequently used for managing *Schizaphis graminum* in the 1990s [31]. Abamectin resistance has been detected in *Plutella xylostella*, *Bemisia tabaci*, and *Tetranychus urticae* [32–34]. Consequently, overcoming insecticide resistance is an ongoing challenge for sustainable pest management.

Despite the significant economic losses caused by *M. dirhodum*, little is known about the sensitivity of *M. dirhodum* to the insecticides commonly used in China to control this pest. In this study, we tested the susceptibility of 19 *M. dirhodum* field populations collected in seven provinces in Northern China in 2017–2019 to seven insecticides from various classes (i.e., neonicotinoids, pyrethroids, organophosphates, and macrolides). A pair-wise correlation analysis revealed some cross-resistance between thiamethoxam and β-cypermethrin. The data presented in this paper provide timely information regarding the insecticide resistance status of *M. dirhodum*, and may be useful for restricting the evolution of insecticide resistance in this important aphid species.

## 2. Materials and Methods

### 2.1. Insect Populations and Insecticides

*Metopolophium dirhodum* samples were collected from wheat fields in Northern China (Figure 1, Table 1). A total of 19 populations were collected from 2017 to 2019 from the following regions: Guide, Qinghai (population code: QIG); Shizuishan, Ningxia (NIS); Yangling, Shaanxi (SHY); Linfen, Shanxi (SHL); Liaocheng, Shandong (SDL); Dezhou, Shandong (SDD); Kashgar prefecture, Xinjiang (XIK); Langfang, Hebei (HLF); Baoding, Hebei (HBD); Dingzhou, Hebei (HDZ); Shijiazhuang, Hebei (HSZ); Xingtai, Hebei (HXT); Handan, Hebei (HHD); and Cangzhou, Hebei (HCZ).

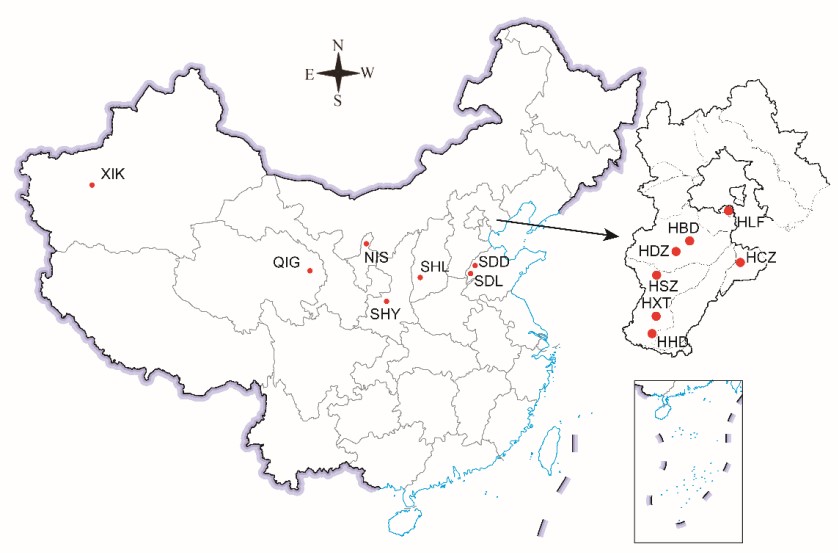

**Figure 1.** *Metopolophium dirhodum* sample collection regions in Northern China.

**Table 1.** Information regarding the collected *Metopolophium dirhodum* field populations.

| No. | Collecting Locality | Code | Collection Date | Longitude and Latitude | HIS [a] | Reference |
|-----|--------------------|------|----------------|------------------------|---------|-----------|
| 1 | Dingzhou, Hebei | HDZ | 17 May 2017 | 38°31′09″ N, 115°04′09″ E | unknown | - |
| 2 | Shijiazhuang, Hebei | HSZ | 17 May 2017 | 37°54′59″ N, 114°45′19″ E | unknown | - |
| 3 | Xingtai, Hebei | HXT | 18 May 2017 | 37°05′04″ N, 114°36′21″ E | unknown | - |
| 4 | Handan, Hebei | HHD | 18 May 2017 | 36°32′31″ N, 114°33′32″ E | unknown | - |
| 5 | Cangzhou, Hebei | HCZ | 26 April 2017 | 38°03′14″ N, 116°40′25″ E | neonicotinoids | [35] |
| 6 | Langfang, Hebei | HLF | 30 April 2017 | 39°30′29″ N, 116°36′09″ E | neonicotinoids | [13] |
| 7 | Baoding, Hebei | HBD | 29 April 2017 | 39°12′56″ N, 115°47′54″ E | unknown | - |
| 8 | Dezhou, Shandong | SDD | 18 May 2017 | 36°57′45″ N, 115°58′01″ E | neonicotinoids pyrethroids organophosphates | [36] |
| 9 | Liaocheng, Shandong | SDL | 18 May 2017 | 36°28′18″ N, 115°39′23″ E | | |
| 10 | Yangling, Shaanxi | SHY | 3 May 2018 | 34°15′33″ N, 108°02′33″ E | neonicotinoids pyrethroids organophosphates macrolides | [37] |
| | | | 13 May 2019 | 34°15′33″ N, 108°02′33″ E | | |
| 11 | Linfen, Shanxi | SHL | 14 May 2018 | 36°06′38″ N, 111°30′04″ E | neonicotinoids pyrethroids organophosphates | [38] |
| | | | 14 May 2019 | 36°06′38″ N, 111°30′04″ E | | |
| 12 | Kashi, Xinjiang | XIK | 7 June 2018 | 38°11′25″ N, 77°11′12″ E | neonicotinoids organophosphates | [39,40] |
| | | | 28 May 2019 | 38°11′25″ N, 77°11′12″ E | | |
| 13 | Shizuishan, Ningxia | NIS | 12 June 2018 | 39°05′57″ N, 106°44′51″ E | pyrethroids organophosphates | [41] |
| | | | 16 June 2019 | 39°05′57″ N, 106°44′51″ E | | |
| 14 | Guide, Qinghai | QIG | 14 June 2018 | 36°02′15″ N, 101°27′13″ E | neonicotinoids pyrethroids organophosphates | [42] |
| | | | 18 June 2019 | 36°02′15″ N, 101°27′13″ E | | |

[a]: the history of insecticide usage.

The following seven insecticides from five different classes were used in this study: 97% thiamethoxam, 96% imidacloprid, 95% beta-cypermethrin, 97% bifenthrin, 95% abamectin, and 97% chlorpyrifos (Beijing Green Agricultural Science and Technology Group Co., Ltd., Beijing, China), and 40% omethoate (emulsifiable) (Hebei Xinxing Chemical Co., Ltd., Baoding, China).

### 2.2. Bioassays

Bioassays were conducted using aphids that were within three generations of being collected from fields. A previously described leaf-dip method [10] was used for the insecticide bioassays. Briefly, the insecticide active ingredients were diluted six or seven times using 0.1% Tween-80 (prepared in water). Wheat leaves containing apterous adult aphids (excluding alatae) were dipped in the diluted insecticide solutions for 3 s. Three replicates of 30–50 aphids were used for each concentration. The mortality rate was calculated for each treatment. Aphids that did not move after being touched by a writing brush were considered dead.

### 2.3. Statistical Analysis

Concentration–mortality data were subjected to a probit analysis, with the data corrected for natural mortality [43]. The median lethal concentration ($LC_{50}$), 95% confidence interval, and slope were calculated using the IBM SPSS program (version 20). The relative resistance ratio (RLR) for each insecticide was calculated on the basis of the median lethal concentration ($LC_{50}$) for the most susceptible field population. The following RLR respectively indicated low, moderate, and high insecticide resistance: RLR $\leq$ 10, 10 < RLR $\leq$ 100, and RLR > 100. Pairwise correlation coefficients for the log $LC_{50}$ values of the field populations treated with imidacloprid, thiamethoxam, beta-cypermethrin, abamectin, and omethoate were calculated according to Pearson's correlation analysis using the SPSS software (IBM Corp., Armonk, NY, USA) to determine the cross-resistance among the insecticides.

## 3. Results

### 3.1. Susceptibility Baseline of M. dirhodum to Seven Insecticides

At present, the baseline values for insecticide resistance in *Metopolophium dirhodum* have not been determined. In this study, the relative resistance ratio for each insecticide was calculated on the basis of the $LC_{50}$ for the most susceptible field population because of the lack of use on a contemporary susceptible reference strain during the bioassay (Table 2).

**Table 2.** Susceptibility baseline of *Metopolophium dirhodum* to seven insecticides.

| Insecticides | Population | N [a] | $LC_{50}$ (95%CI; mg/L) [b] | Slope $\pm$ SE | *p*-Value | Correlation Coefficient |
|---|---|---|---|---|---|---|
| Thiamethoxam | HBD-2017 | 1849 | 4.27 (2.27–8.03) | 4.64 $\pm$ 0.10 | 0.0001 | 0.9810 |
| Imidacloprid | SDD-2017 | 1817 | 3.93 (2.55–6.05) | 4.71 $\pm$ 0.05 | 0.0001 | 0.9940 |
| Beta-cypermethrin | NIS-2019 | 713 | 0.52 (0.48–0.56) | 0.58 $\pm$ 0.05 | 0.0001 | 0.9999 |
| Omethoate | SHL-2019 | 460 | 18.63 (8.44–41.03) | 0.75 $\pm$ 0.12 | 0.0087 | 0.9625 |
| Bifenthrin | QIG-2019 | 543 | 9.47 (5.68–15.80) | 0.57 $\pm$ 0.05 | 0.0017 | 0.9874 |
| Chlorpyrifos | SHY-2019 | 702 | 0.44 (0.31–4066) | 2.33 $\pm$ 0.37 | 0.0083 | 0.9637 |
| Abamectin | HHD-2019 | 2109 | 1.60 (0.97–2.63) | 4.80 $\pm$ 0.12 | 0.0014 | 0.9697 |

[a] Number of tested aphids. [b] Median lethal concentration and 95% confidence interval.

### 3.2. Monitoring Sensitivity to Seven Insecticides in Northern China

The relative resistance levels varied among the field populations collected from various locations in Northern China (Figure 2). For the neonicotinoids, the NIS-2018 and SHY-2018 populations were highly resistant to thiamethoxam, with an RLR of 134.03 and 103.30, respectively. Additionally, 16 populations were ranging from susceptible to showing moderate levels of resistance to thiamethoxam, with an RLR of 2.04–46.77, whereas only four populations were moderately resistant to imidacloprid, with an RLR of 10.95–57.29 and 14 populations ranging from susceptible to showing tolerance or low levels of re-

sistance, with an RLR of 1.50–5.56. (Table 3). The bioassay results for the pyrethroids indicated that the NIS-2018 population was highly resistant to beta-cypermethrin, with an RLR of 121.42, but 15 populations were ranging from susceptible to showing moderate levels of resistance, with an RLR of 2.40–96.75. However, the tested populations were equally susceptible to bifenthrin. Regarding the susceptibility to the older generation organophosphate insecticides, the SHL-2018 and NIS-2019 populations were moderately resistant to omethoate, with an RR of 18.73 and 11.82, respectively. An analysis of the susceptibility to chlorpyrifos revealed that the tested populations were equally susceptible (Table 4). Moreover, 18 populations were ranging from susceptible to showing moderate levels of resistance to abamectin, with an RLR of 1.10–25.89 (Table 5).

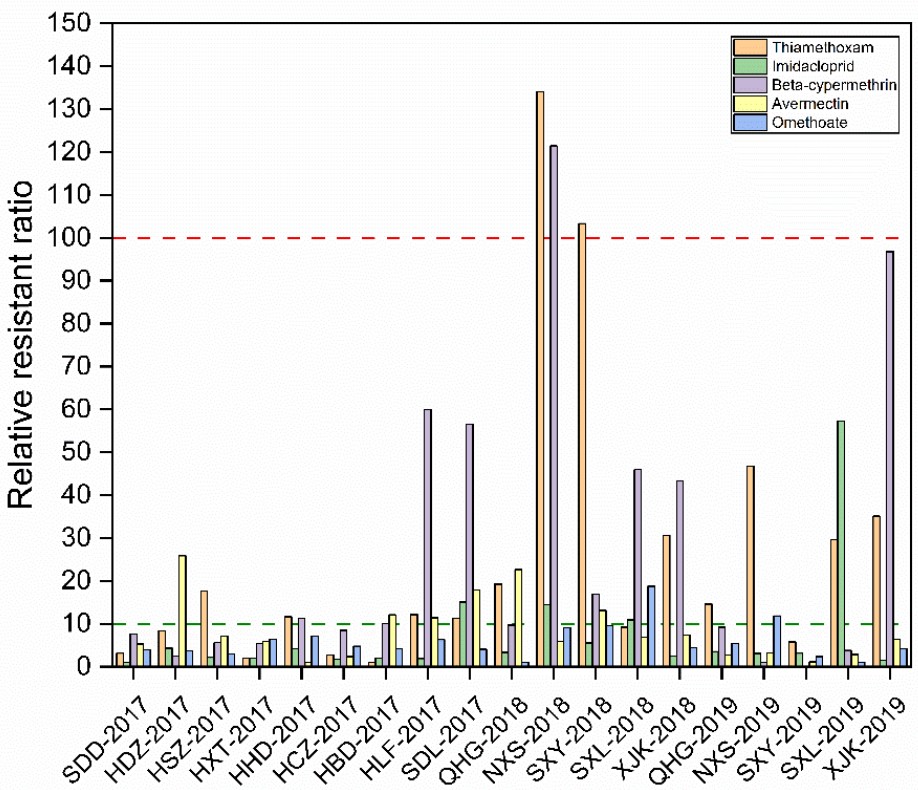

**Figure 2.** Relative resistance levels of *Metopolophium dirhodum* field populations.

**Table 3.** Susceptibility of Metopolophium dirhodum field populations to imidacloprid and thiamethoxam.

| Population | N [a] | Thiamethoxam | | | | | Imidacloprid | | | | | |
| | | LC$_{50}$ (95% CI; mg/L) [b] | Slope ± SE | *p*-Value | Correlation Coefficient | RLR [c] | N | LC$_{50}$ (95% CI; mg/L) | Slope ± SE | *p*-Value | Correlation Coefficient | RLR [c] |
|---|---|---|---|---|---|---|---|---|---|---|---|---|
| SDD-2017 | 1857 | 13.42 (7.42–24.95) | 4.23 ± 0.13 | 0.0009 | 0.9753 | 3.14 | 1817 | 3.93 (2.55–6.05) | 4.71 ± 0.05 | 0.0001 | 0.9940 | 1.00 |
| HDZ-2017 | 2302 | 35.60 (25.80–49.10) | 3.76 ± 0.10 | 0.0001 | 0.9926 | 8.34 | 1303 | 16.97 (12.38–23.27) | 4.20 ± 0.06 | 0.0001 | 0.9960 | 4.32 |
| HSZ-2017 | 843 | 75.25 (58.19–97.31) | 3.31 ± 0.13 | 0.0006 | 0.9934 | 17.62 | 2212 | 8.51 (5.31–13.63) | 4.38 ± 0.09 | 0.0009 | 0.9919 | 2.17 |
| HXT-2017 | 1951 | 8.71 (6.58–11.54) | 4.21 ± 0.08 | 0.0001 | 0.9939 | 2.04 | 1578 | 8.07 2.63–24.75) | 4.46 ± 0.20 | 0.0177 | 0.9543 | 2.05 |
| HHD-2017 | 1746 | 49.56 (7.22–16.59) | 3.99 ± 0.14 | 0.0001 | 0.9780 | 11.61 | 1742 | 16.72 (11.55–24.21) | 4.02 ± 0.08 | 0.0004 | 0.9951 | 4.25 |
| HCZ-2017 | 1968 | 11.39 (7.04–18.43) | 4.38 ± 0.10 | 0.0001 | 0.9839 | 2.67 | 1562 | 6.86 (4.00–11.78) | 4.46 ± 0.08 | 0.0001 | 0.9907 | 1.75 |
| HBD-2017 | 1849 | 4.27 (2.27–8.03) | 4.64 ± 0.10 | 0.0001 | 0.9810 | 1.00 | 2692 | 7.72 (3.97–14.93) | 4.41 ± 0.12 | 0.0004 | 0.9840 | 1.96 |
| HLF-2017 | 1486 | 51.81 (55.28–80.00) | 4.04 ± 0.03 | 0.0001 | 0.9716 | 12.13 | 1430 | 7.55 (3.99–14.30) | 4.45 ± 0.13 | 0.0001 | 0.9677 | 1.92 |
| SDL-2017 | 1726 | 48.10 (31.28–73.95) | 3.54 ± 0.14 | 0.0005 | 0.9826 | 11.26 | 1236 | 59.24 (35.46–98.98) | 3.63 ± 0.23 | 0.0008 | 0.9561 | 15.07 |
| QIG-2018 | 2426 | 82.33 (29.86–227.01) | 0.48 ± 0.12 | 0.0293 | 0.9151 | 19.28 | 925 | 13.09 (9.37–18.29) | 4.03 ± 0.11 | 0.0005 | 0.9943 | 3.33 |
| NIS-2018 | 550 | 572.29 (281.45–1163.67) | 0.78 ± 0.10 | 0.0045 | 0.9759 | 134.03 | 654 | 56.63 (14.41–222.55) | 3.79 ± 0.49 | 0.0594 | 0.8634 | 14.41 |
| SHY-2018 | 766 | 441.11 (150.42–1293.56) | 0.53 ± 0.11 | 0.0173 | 0.9404 | 103.30 | 511 | 21.87 (11.14–42.95) | 4.39 ± 0.11 | 0.0070 | 0.9674 | 5.56 |
| SHL-2018 | 2289 | 39.32 (25.96–59.56) | 0.62 ± 0.06 | 0.0020 | 0.9860 | 9.22 | 1145 | 43.05 (38.63–47.97) | 4.02 ± 0.03 | 0.0005 | 0.9995 | 10.95 |
| XIK-2018 | 560 | 130.48 (100.43–169.51) | 0.86 ± 0.06 | 0.0006 | 0.9939 | 30.56 | 913 | 9.51 (3.71–24.40) | 4.43 ± 0.20 | 0.0085 | 0.9630 | 2.42 |
| QIG-2019 | 780 | 62.29 (44.18–87.82) | 0.77 ± 0.06 | 0.0010 | 0.9913 | 14.59 | 686 | 13.96 (5.07–38.44) | 0.99 ± 0.20 | 0.0150 | 0.9459 | 3.54 |
| NIS-2019 | 689 | 199.72 (120.76–330.31) | 0.88 ± 0.09 | 0.0019 | 0.9864 | 46.77 | 515 | 12.23 (8.24–18.15) | 0.59 ± 0.06 | 0.0111 | 0.9889 | 3.11 |
| SHY-2019 | 546 | 24.58 (10.99–55.00) | 0.40 ± 0.07 | 0.0116 | 0.9545 | 5.76 | 603 | 12.29 (9.41–16.05) | 0.80 ± 0.05 | 0.0004 | 0.9951 | 3.13 |
| SHL-2019 | 660 | 126.19 (56.44–282.15) | 0.68 ± 0.10 | 0.0065 | 0.9691 | 29.55 | 556 | 225.16 (114.66–442.15) | 1.02 ± 0.24 | 0.1494 | 0.9726 | 57.29 |
| XIK-2019 | 571 | 149.86 (62.38–360.01) | 1.24 ± 0.22 | 0.0110 | 0.9561 | 35.10 | 605 | 5.89 (2.42–14.33) | 0.23 ± 0.04 | 0.0229 | 0.9771 | 1.50 |

[a] Number of aphids used in bioassays. [b] Median lethal concentration and 95% confidence interval. [c] Relative resistance ratio.

**Table 4.** Susceptibility of *Metopolophium dirhodum* field populations to beta-cypermethrin, bifenthrin, omethoate, and chlorpyrifos.

| Population | N [a] | Beta-Cypermethrin | | | | | N | Omethoate | | | | |
|---|---|---|---|---|---|---|---|---|---|---|---|---|
| | | LC$_{50}$ (95% CI; mg/L) [b] | Slope ± SE | *p*-Value | Correlation Coefficient | RLR [c] | | LC$_{50}$ (95% CI; mg/L) | Slope ± SE | *p*-Value | Correlation Coefficient | RLR |
| SDD-2017 | 1532 | 3.96 (1.88–8.33) | 4.55 ± 0.14 | 0.0017 | 0.9664 | 7.62 | 2129 | 73.22 (45.68–117.34) | 2.58 ± 0.23 | 0.0025 | 0.9835 | 3.93 |
| HDZ-2017 | 1272 | 1.25 (0.64–2.46) | 4.91 ± 0.14 | 0.0008 | 0.9764 | 2.40 | 3266 | 68.44 (47.45–98.71) | 2.03 ± 0.38 | 0.0026 | 0.9831 | 3.67 |
| HSZ-2017 | 1875 | 2.92 (15.30–45.69) | 4.21 ± 0.12 | 0.0002 | 0.9971 | 5.62 | 1228 | 54.49 (82.35–1159.44) | 3.21 ± 0.31 | 0.0001 | 0.9840 | 2.92 |
| HXT-2017 | 2844 | 2.85 (1.83–4.42) | 4.6 ± 0.10 | 0.0001 | 0.9818 | 5.48 | 2842 | 119.91 (96.03–149.72) | 1.45 ± 0.23 | 0.0008 | 0.9922 | 6.44 |
| HHD-2017 | 1733 | 5.84 (3.26–10.47) | 4.43 ± 0.13 | 0.0003 | 0.9704 | 11.23 | 1618 | 133.40 (89.28–199.32) | 1.09 ± 0.76 | 0.0103 | 0.9575 | 7.16 |
| HCZ-2017 | 1781 | 4.42 (2.56–7.63) | 4.47 ± 0.12 | 0.0007 | 0.9788 | 8.50 | 2076 | 88.43 (77.17–101.34) | 2.17 ± 0.13 | 0.0002 | 0.9974 | 4.75 |
| HBD-2017 | 1824 | 5.20 (2.15–12.63) | 4.41 ± 0.20 | 0.0075 | 0.9661 | 10.00 | 2781 | 79.11 (64.85–96.51) | 2.09 ± 0.24 | 0.0001 | 0.9905 | 4.25 |
| HLF-2017 | 1954 | 31.18 (18.33–53.03) | 4.08 ± 0.13 | 0.0002 | 0.9731 | 59.96 | 3001 | 117.61 (95.96–144.14) | 2.02 ± 0.19 | 0.0001 | 0.9941 | 6.31 |
| SDL-2017 | 1355 | 29.40 (19.49–44.33) | 3.64 ± 0.16 | 0.0006 | 0.9792 | 56.54 | 1290 | 74.97 (59.80–93.98) | 0.78 ± 0.26 | 0.0012 | 0.9898 | 4.02 |
| QIG-2018 | 1071 | 5.05(2.21–11.52) | 0.42 ± 0.06 | 0.0071 | 0.9673 | 9.71 | 809 | 19.90 (16.31–24.29) | 0.99 ± 0.04 | 0.0002 | 0.9971 | 1.07 |
| NIS-2018 | 475 | 63.14 (39.15–101.80) | 0.93 ± 0.09 | 0.0021 | 0.9856 | 121.42 | 309 | 169.53 (39.13–734.52) | 0.66 ± 0.17 | 0.0285 | 0.9168 | 9.10 |
| SHY-2018 | 702 | 8.81 (0.97–80.43) | 0.53 ± 0.21 | 0.0849 | 0.8259 | 16.94 | 666 | 177.05 (63.99–489.87) | 1.29 ± 0.49 | 0.2323 | 0.9342 | 9.50 |
| SHL-2018 | 1644 | 23.90 (16.10–35.50) | 0.85 ± 0.08 | 0.0015 | 0.9884 | 45.96 | 1791 | 348.86 (67.74–489.87) | 0.49 ± 0.12 | 0.0252 | 0.9342 | 18.73 |
| XIK-2018 | 620 | 22.55 (15.16–33.57) | 0.76 ± 0.08 | 0.0109 | 0.9891 | 43.37 | 499 | 82.63 (30.19–226.18) | 1.07 ± 0.21 | 0.0154 | 0.9450 | 4.44 |
| QIG-2019 | 496 | 4.79 (1.75–13.11) | 0.68 ± 0.15 | 0.0439 | 0.9561 | 9.21 | 724 | 100.70 (44.77–226.52) | 1.07 ± 0.19 | 0.0104 | 0.9576 | 5.41 |
| NIS-2019 | 713 | 0.52 (0.48–0.56) | 0.58 ± 0.05 | 0.0001 | 0.9999 | 1.00 | 520 | 220.23 (167.30–289.92) | 1.68 ± 0.18 | 0.0674 | 0.9944 | 11.82 |
| SHY-2019 | | | | | | ND [d] | 723 | 43.80 (30.85–62.17) | 1.55 ± 0.14 | 0.0085 | 0.9915 | 2.35 |
| SHL-2019 | 714 | 1.96 (0.59–6.55) | 0.53 ± 0.09 | 0.0115 | 0.9546 | 3.77 | 460 | 18.63 (8.44–41.03) | 0.75 ± 0.12 | 0.0087 | 0.9625 | 1.00 |
| XIK-2019 | 487 | 50.31 (29.09–86.98) | 0.85 ± 0.13 | 0.0232 | 0.9768 | 96.75 | 439 | 78.06 (35.90–169.75) | 1.18 ± 0.26 | 0.0438 | 0.9562 | 4.19 |

| Population | N [a] | Bifenthrin | | | | | N | Chlorpyrifos | | | | |
|---|---|---|---|---|---|---|---|---|---|---|---|---|
| | | LC$_{50}$ (95% CI; mg/L) [b] | Slope ± SE | *p*-Value | Correlation Coefficient | RLR [c] | | LC$_{50}$ (95% CI; mg/L) | Slope ± SE | *p*-Value | Correlation Coefficient | RLR |
| QIG-2019 | 543 | 9.47 (5.68–15.80) | 0.57 ± 0.05 | 0.0017 | 0.9874 | 1.00 | 702 | 0.49 (0.30–0.82) | 4.72 ± 1.12 | 0.0247 | 0.9244 | 1.11 |
| NIS-2019 | 571 | 10.09 (8.50–11.99) | 0.65 ± 0.02 | 0.0001 | 0.9981 | 1.14 | 704 | 0.80 (0.66–0.96) | 3.91 ± 0.45 | 0.0130 | 0.9870 | 1.82 |
| SHY-2019 | | | | | | ND [d] | 702 | 0.44 (0.31–0.63) | 2.33 ± 0.37 | 0.0083 | 0.9637 | 1.00 |
| SHL-2019 | 606 | 12.78 (5.84–27.98) | 0.69 ± 0.12 | 0.0102 | 0.9582 | 1.21 | 712 | 1.32 (1.21–1.44) | 3.09 ± 0.14 | 0.0002 | 0.9970 | 3.09 |
| XIK-2019 | 409 | 57.01 (25.86–125.69) | 0.81 ± 0.21 | 0.0302 | 0.9134 | 6.02 | 784 | 3.34 (1.40–7.97) | 1.61 ± 0.33 | 0.0396 | 0.9604 | 7.59 |

[a] Number of tested aphids. [b] Median lethal concentration and 95% confidence interval. [c] Relative resistance ratio. [d] No data (i.e., not tested).

**Table 5.** Susceptibility of *Metopolophium dirhodum* field populations to abamectin.

| Population | N [a] | LC$_{50}$ (95%CI; mg/L) [b] | Slope ± SE | *p*-Value | Correlation Coefficient | RLR [c] |
|---|---|---|---|---|---|---|
| HHD-2017 | 2109 | 1.60 (0.97–2.63) | 4.80 ± 0.12 | 0.0014 | 0.9697 | 1.00 |
| HDZ-2017 | 1399 | 41.42 (23.49–73.04) | 3.46 ± 0.19 | 0.0056 | 0.9719 | 25.89 |
| HSZ-2017 | 1722 | 11.35 (7.17–17.98) | 3.72 ± 0.18 | 0.0050 | 0.9741 | 7.09 |
| HXT-2017 | 1740 | 9.44 (6.13–14.55) | 4.18 ± 0.12 | 0.0001 | 0.9805 | 5.90 |
| SDD-2017 | 2189 | 8.46 (5.62–12.74) | 3.43 ± 0.24 | 0.0009 | 0.9752 | 5.29 |
| HCZ-2017 | 1918 | 3.82 (2.34–6.24) | 4.09 ± 0.19 | 0.0010 | 0.9735 | 2.39 |
| HBD-2017 | 2245 | 19.35 (13.91–26.90) | 3.91 ± 0.08 | 0.0005 | 0.9941 | 12.09 |
| HLF-2017 | 1258 | 18.22 (12.70–26.14) | 4.00 ± 0.00 | 0.0021 | 0.9852 | 11.39 |
| SDL-2017 | 1103 | 28.57 (20.78–39.29) | 2.39 ± 0.36 | 0.0014 | 0.9695 | 17.86 |
| QIG-2018 | 1488 | 36.19 (17.33–75.59) | 0.92 ± 0.15 | 0.0087 | 0.9624 | 22.62 |
| NIS-2018 | 771 | 9.52 (2.69–33.60) | 0.81 ± 0.21 | 0.0311 | 0.9116 | 5.95 |
| SHY-2018 | 621 | 21.01 (10.13–43.6) | 1.18 ± 0.19 | 0.0087 | 0.9623 | 13.13 |
| SHL-2018 | 1847 | 11.03 (4.36–27.89) | 0.67 ± 0.13 | 0.0145 | 0.9823 | 6.89 |
| XIK-2018 | 696 | 11.88 (9.24–15.29) | 1.07 ± 0.06 | 0.0003 | 0.9957 | 7.43 |
| QIG-2019 | 580 | 4.28 (2.57–7.12) | 0.97 ± 0.09 | 0.0016 | 0.9880 | 2.68 |
| NIS-2019 | 676 | 5.23 (4.48–6.09) | 1.78 ± 0.09 | 0.0024 | 0.9976 | 3.27 |
| SHY-2019 | 750 | 1.76 (0.88–3.49) | 0.81 ± 0.10 | 0.0035 | 0.9795 | 1.10 |
| SHL-2019 | 576 | 4.53 (3.52–5.84) | 1.49 ± 0.12 | 0.0060 | 0.9940 | 2.83 |
| XIK-2019 | 687 | 10.20 (7.23–14.40) | 1.04 ± 0.07 | 0.0008 | 0.9923 | 6.38 |

[a] Number of aphids used in bioassays. [b] Median lethal concentration and 95% confidence interval. [c] Relative resistance ratio.

### 3.3. Insecticide Resistance at Five Locations in 2018–2019

The insecticide susceptibility of *M. dirhodum* collected from the same region in 2018 and 2019 was analyzed (Figure 3). Our results indicated that the sensitivity of *M. dirhodum* to five tested insecticides fluctuated between 2018 and 2019. More specifically, although the 2-year analysis revealed similar resistance levels in most regions, substantial differences between years were detected for the resistance to thiamethoxam in NIS and SHY (Figure 3B), the resistance to beta-cypermethrin in NIS and SHL (Figure 3C), the resistance to omethoate in SHL (Figure 3D), and the resistance to abamectin in QIG and SHY (Figure 3E).

Pair-wise correlation between the log LC$_{50}$ values of different insecticides.

There were no significant correlations among the evaluated insecticides, with the exception of a significant positive correlation between thiamethoxam and beta-cypermethrin (Table 6).

**Table 6.** Pair-wise correlation analysis of the LC$_{50}$ values of five insecticides for 19 *Metopolophium dirhodum* field populations.

| Insecticides | Imidacloprid | Thiamethoxam | β-Cypermethrin | Abamectin |
|---|---|---|---|---|
| Thiamethoxam | 0.165 | | | |
| β-cypermethrin | 0.023 | 0.504 * | | |
| Abamectin | −0.119 | −0.041 | 0.024 | |
| Omethoate | −0.132 | 0.283 | 0.247 | −0.193 |

* Positive correlation between LC$_{50}$ values (0.05 significance level).

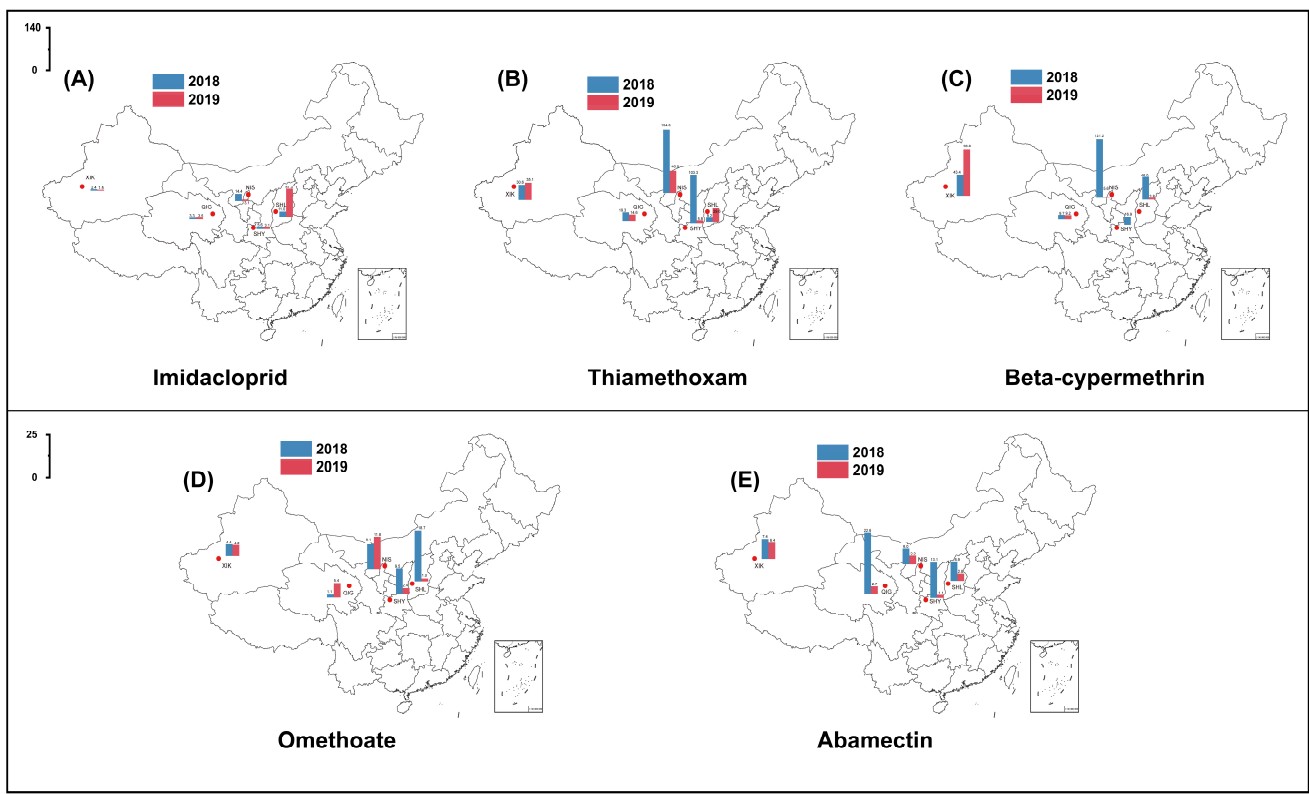

**Figure 3.** Insecticide susceptibility of *Metopolophium dirhodum* collected at the same sites in 2018 and 2019, (**A**) Imidacloprid; (**B**) Thiamethoxam; (**C**) Beta-cypermethrin; (**D**) Omethoate; (**E**) Abamectin. Blue and red bars represent the resistance ratios in 2018 and 2019, respectively.

## 4. Discussion

Previous research determined the baseline toxicities of insecticides used to control wheat aphids as well as the corresponding susceptibility levels [15,20,21]. In the current study, we analyzed the susceptibility of the wheat aphid *M. dirhodum* to insecticides by comparing the lethal concentrations of various insecticides for 19 field populations. We also evaluated the resistance of *M. dirhodum* collected from 14 regions in Northern China to seven commonly used insecticides and determined the likelihood of cross-resistance among five insecticides.

Our results indicated that some *M. dirhodum* populations (4 of 19) exhibited moderate levels of resistance to imidacloprid. Additionally, two populations (NIS-2018 and SHY-2018) were highly resistant to thiamethoxam. An earlier investigation assessing the effect of coating seeds with imidacloprid on laboratory populations of four wheat aphids confirmed that the seed treatment could effectively control *R. padi*, *S. avenae*, and *S. graminum*, but not *M. dirhodum* [13]. This is particularly important considering *M. dirhodum* may overtake *R. padi* and *S. avenae* as the primary aphid species on wheat plants derived from neonicotinoid-treated seeds [10]. Furthermore, different wheat aphids often harm crops at the same time. Consequently, the effect of neonicotinoid seed treatments on the prevention and control of *M. dirhodum* infestations in the field will need to be investigated. The data presented herein may be useful for optimizing the use of neonicotinoids and decreasing the environmental effects associated with the application of multiple pesticides.

Regarding pyrethroids, the limited *M. dirhodum* populations collected in 2019 exhibited low or no resistance to bifenthrin. In contrast, higher levels of resistance to beta-cypermethrin were detected, including one highly resistant population and moderately resistant populations. Clearly, determining the likelihood an insecticide will fail to control *M. dirhodum* is warranted. In Argentina, the most widely used insecticides for controlling

*M. dirhodum* are chlorpyrifos, dimethoate, and pirimicarb [44]. Organophosphates were used to control *M. dirhodum* for a long time, but they are rarely used now [45]. Moreover, because organophosphates effectively minimized damages caused by *M. dirhodum*, there has been relatively little attention paid to this aphid species. The bioassay results of the current study, in which *M. dirhodum* was susceptible to organophosphates, were as expected. Additionally, *M. dirhodum* was also relatively susceptible to abamectin.

To improve the prevention and control of *M. dirhodum* infestations, we examined the resistance patterns among pesticides to identify the best compounds for managing *M. dirhodum*. Significant cross-resistance was detected between β-cypermethrin and thiamethoxam. A thiamethoxam-resistant cotton aphid strain reportedly developed increasing resistance to several pyrethroids [46]. A recent study proved that the resistance to λ-cyhalothrin is related to the resistance to thiamethoxam in *R. padi* [47]. Some cytochrome P450 monooxygenase genes encoding CYP6 and CYP3 family members are associated with the resistance to both neonicotinoids and pyrethroids [48,49]. Furthermore, there is evidence that NADPH–cytochrome P450 contributes to the resistance to β-cypermethrin and imidacloprid in *N. lugens* [50]. Despite the intensive use of chemical treatments, *M. dirhodum* has still managed to invade new areas and damage crops in northern China [13]. Previously or seldomly used insecticides are still effective for managing pests, including pyrethroids, organophosphates, sulfoximines (e.g., sulfoxaflor), and macrolides (e.g., abamectin).

## 5. Conclusions

Our results suggest that *M. dirhodum* field populations are resistant to the neonicotinoid and pyrethroid insecticides most frequently used to control this aphid species in SHY (Yangling, Shaanxi) and NIS (Shizuishan, Ningxia) of China. The efficacy of neonicotinoid seed treatments for controlling *M. dirhodum* should be evaluated in future studies. Our bioassay results indicate that abamectin and bifenthrin are effective against *M. dirhodum* and can be used as alternatives to insecticides to which *M. dirhodum* has evolved high levels of resistance. Unfortunately, rotating the use of thiamethoxam and beta-cypermethrin to control *M. dirhodum* in parts of China should probably be limited. Overall, we recommend that monitoring and control programs should be strengthened in regions where *M. dirhodum* is distributed.

**Author Contributions:** Conceptualization, X.Z., H.G. and X.Y.; methodology, X.Z. and P.G.; software, P.G.; formal analysis, P.G. and X.L. (Xinan Li); investigation P.G., X.L. (Xinan Li), C.W., S.Z. and Q.L.; writing—original draft preparation, P.G.; writing—review and editing, E.L., G.L., X.Z., Y.Z. and X.L. (Xiangrui Li). All authors have read and agreed to the published version of the manuscript.

**Funding:** This research was funded by the National Key Research and Development Program of China (2017YFD0200303), China Agriculture Research System of MOF and MARA (CARS-3), and the Training Program for outstanding young talents in science and technology of Xinjiang Uygur Autonomous Region (2018Q041), SINOGRAIN II (CHN-17/0019): Technological Innovation to Support Environmentally-Friendly Food Production and Food Safety Under a Changing Climate-Opportunities and Challenges for Norway-China Cooperation.

**Institutional Review Board Statement:** Not applicable.

**Informed Consent Statement:** Not applicable.

**Data Availability Statement:** Most of the recorded data are available in all Tables in the manuscript.

**Acknowledgments:** We thank Liwen Bianji for editing the English text of a draft of this manuscript.

**Conflicts of Interest:** The authors declare no conflict of interest.

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
