# Peer review of "The Sensitivity of Field Populations of Metopolophium dirhodum (Walker) (Hemiptera: Aphididae) to Seven Insecticides in Northern China"

_agronomy, doi:10.3390/agronomy11081556_

Round 1
Reviewer 1 Report
This is a useful study but changes are recommended, in particular regarding the calculation of RR values (see below).
Title
It is not clear from Figure 1 why the title includes 'Northwest'? One population tested (XIK) is clearly in the NW of China but the other populations appear to be north/central China?
'Northwest' also appears in the Abstract, Introduction, Materials and Methods and Results.
The Discussion refers to 'northern' China, which would seem more appropriate?
Abstract
Lines: 29-31: populations with RR of <10 may be showing tolerance rather than resistance; the latter being confirmed by additional testing which was not done in this work. Here there is inconsistency in the use of the terms 'low resistance' and 'susceptible'. Populations with RR of 1.50 - 57.39 would be better described as 'ranging from susceptible to showing moderate levels of resistance'; and with RR 1.14 - 6.02 and 1.11 - 7.59 as 'ranging from susceptible to showing tolerance or low levels of resistance'.
Line 58: this sentence needs qualifying. Insecticides are not 'an indispensable component of pest management programmes'; suggest...'Insecticides remain an important component of many pest management programmes.'
Lines 67-68: suggest change part of sentence on neonicotinoids to...'........insecticides currently used against sap-feeding insect pests, although their usage has been banned or restricted in Europe, and restricted in some states in the USA, due to concerns about side effects on bees and other pollinators.' [add REFS]
Line 73: suggest change...'common use'... to .'..long-term use...'
Methods and Materials
Please include, where possible, the history of insecticide usage [including the seven compounds tested or same class of compound] for the different regions where the 19 aphid populations were collected. This could be included, for example, within Table 1, using abbreviations for each compound/chemical class?
It is unfortunate that a control, insecticide susceptible population was not used in the present study. See comments in Results below.
Results
Lines 128-129: the first sentence notes that 'the baseline values for insecticide resistance in M. dirhodum have not been determined'
Lines 129-132: the second sentence states that the relative resistance (RR) ratio for each compound [therefore] used the most susceptible of the field populations due to the lack of a 'contemporary susceptible reference strain during the bioassay'... The bioassay results meant that a different field population was used for each insecticide (Table 2) and the estimations of 'relative' resistance ratios for different insecticides cannot be compared within any individual population.
It is also apparent that at least some of the baseline populations used were likely to be resistant to some of the other compounds; e.g. NIS 2019, the baseline population for beta-cypermethrin, appears to have high levels of resistance to the two neonicotinoids tested.
The same bioassay method was used as in a previous study from this laboratory [Ref 10], in which a laboratory, insecticide susceptible population of M. dirhodum was used in assays against five of the seven insecticides tested in the present study.
1) Couldn't this population or a similar one be used to generate baseline data against the seven compounds used here?
2) If there is a strong reason why this is not possible, the authors could indicate what the effect on RR values would be for five of the compounds if their respective LC50 values for each of the 19 populations in the present study were compared against the susceptible population's LC50 data in Ref 10?
Lines 137-150 and 169-179: please check usage of 'susceptible', 'moderate resistance' etc [see comment above on lines 29-31].
The text for this section should also be revised in view of comments above regarding the problem of using different baseline populations for different insecticides and the solutions mentioned.
Discussion
Text to be modified based on changes to Results text.
Conclusions
Line 231: suggest change ....'We confirmed'... to.. 'Our results suggest..........most frequently used to control this aphid species in ..add regions(s).. China.'
Author Response
Review 1
Title
It is not clear from Figure 1 why the title includes 'Northwest'? One population tested (XIK) is clearly in the NW of China but the other populations appear to be north/central China? 'Northwest' also appears in the Abstract, Introduction, Materials and Methods and Results. The Discussion refers to 'northern' China, which would seem more appropriate?
Author’ s Response: We changed ‘northwest’ to ‘northern’ in whole paper.
Abstract
Lines: 29-31: populations with RR of <10 may be showing tolerance rather than resistance; the latter being confirmed by additional testing which was not done in this work. Here there is inconsistency in the use of the terms 'low resistance' and 'susceptible'. Populations with RR of 1.50 - 57.39 would be better described as 'ranging from susceptible to showing moderate levels of resistance'; and with RR 1.14 - 6.02 and 1.11 - 7.59 as 'ranging from susceptible to showing tolerance or low levels of resistance'.
Author’ s Response: Revised (Lines 28-32).
Line 58: this sentence needs qualifying. Insecticides are not 'an indispensable component of pest management programs'; suggest...'Insecticides remain an important component of many pest management programs.'
Author’ s Response: Revised (Line 59).
Lines 67-68: suggest change part of sentence on neonicotinoids to...'........insecticides currently used against sap-feeding insect pests, although their usage has been banned or restricted in Europe, and restricted in some states in the USA, due to concerns about side effects on bees and other pollinators.' [add REFS]
Author’ s Response: Revised (Lines 68-70).
Line 73: suggest change...'common use'... to 'long-term use...'
Author’ s Response: Revised (Lines 74-75).
Methods and Materials
Please include, where possible, the history of insecticide usage [including the seven compounds tested or same class of compound] for the different regions where the 19 aphid populations were collected. This could be included, for example, within Table 1, using abbreviations for each compound/chemical class? It is unfortunate that a control, insecticide susceptible population was not used in the present study. See comments in Results below.
Author’ s Response: Revised (Table 1).
Results
Lines 128-129: the first sentence notes that 'the baseline values for insecticide resistance in M. dirhodum have not been determined' Lines 129-132: the second sentence states that the relative resistance (RR) ratio for each compound [therefore] used the most susceptible of the field populations due to the lack of a 'contemporary susceptible reference strain during the bioassay'... The bioassay results meant that a different field population was used for each insecticide (Table 2) and the estimations of 'relative' resistance ratios for different insecticides cannot be compared within any individual population. It is also apparent that at least some of the baseline populations used were likely to be resistant to some of the other compounds; e.g. NIS 2019, the baseline population for beta-cypermethrin, appears to have high levels of resistance to the two neonicotinoids tested. The same bioassay method was used as in a previous study from this laboratory [Ref 10], in which a laboratory, insecticide susceptible population of M. dirhodum was used in assays against five of the seven insecticides tested in the present study.
1) Couldn't this population or a similar one be used to generate baseline data against the seven compounds used here?
Author’ s Response: In all field populations, no particular population was found to be sensitive to all the insecticides tested, so each insecticide was used with the corresponding sensitive population to calculate the relative resistance ratio to evaluate the sensitivity of different field populations to a particular agent. We believe that this is closer to the situation in the field.
In a previous study [Ref 10], we compared the susceptibility of four species of wheat aphids from the same region to seven insecticides and found that the M. dirhodum was more resistant to neonicotinoids. Despite being reared indoors for a long time without exposure to pesticides, it still had a higher LC50 compared to other field populations collected later. Our lab is returning a broad spectrum of susceptible population.
2) If there is a strong reason why this is not possible, the authors could indicate what the effect on RR values would be for five of the compounds if their respective LC50 values for each of the 19 populations in the present study were compared against the susceptible population's LC50 data in Ref 10?
Author’ s Response: The population of M. dirhodum in [Ref 10] was collected from a trial field where imidacloprid was used as a long-term seed treatment. It was not a susceptible population in this study. If respective LC50 values for each of the 19 populations were compared against the LC50 data in Ref 10 more than half of the 19 populations will have lower RR values. It would give the false impression that these populations are not resistant to insecticides, which is inconsistent with the facts.
Lines 137-150 and 169-179: please check usage of 'susceptible', 'moderate resistance' etc [see comment above on lines 29-31.
Author’ s Response: Revised (Lines 143-156).
The text for this section should also be revised in view of comments above regarding the problem of using different baseline populations for different insecticides and the solutions mentioned.
Author’ s Response: Revised.
Discussion
Text to be modified based on changes to Results text.
Author’ s Response: Done.
Conclusions
Line 231: suggest change ....'We confirmed'... to. 'Our results suggest..........most frequently used to control this aphid species in.add regions(s).. China.'
Author’ s Response: Revised (Lines 237-239).

Reviewer 2 Report
The invasive pest insect Metopolophium dirhodum has become a world-wide pest in recent year. The dada shown in this manuscript as to insecticidal resistant against different types of various chemicals of M. dirhodum field populations collected from the whole extent of China are valuable information for development of future research in effective control of this pest. I appreciate the effort of your field works. I cannot find any major revisions in this manuscript.
Author Response
Thank you very much for your recognition of our work